# DNA Barcoding, Phylogenetic Analysis and Secondary Structure Predictions of *Nepenthes ampullaria, Nepenthes gracilis* and *Nepenthes rafflesiana*

**DOI:** 10.3390/genes14030697

**Published:** 2023-03-11

**Authors:** Nur Azreen Saidon, Alina Wagiran, Abdul Fatah A. Samad, Faezah Mohd Salleh, Farhan Mohamed, Jaeyres Jani, Alona C Linatoc

**Affiliations:** 1Department of Biosciences, Faculty of Science, Universiti Teknologi Malaysia, Johor Bahru 81310, Johor, Malaysia; 2Media and Games Innovation Centre, Universiti Teknologi Malaysia, Johor Bahru 81310, Johor, Malaysia; 3Borneo Medical and Health Research Center, Faculty of Medicine and Health Sciences, Universiti Malaysia Sabah, Kota Kinabalu 88400, Sabah, Malaysia; 4Faculty of Applied Sciences & Technology, Universiti Tun Hussein Onn Malaysia (UTHM), Hab Pendidikan Tinggi Pagoh, KM1, Jalan Panchor, Muar 84600, Johor, Malaysia; 5College of Forestry and Natural Resources, University of the Philippines Los Banos College, Los Baños 4031, Laguna, Philippines

**Keywords:** DNA barcode, *Nepenthes*, *rbc*L, ITS1, ITS2, phylogenetic, barcoding gap, secondary structure predictions

## Abstract

Nepentheceae, the most prominent carnivorous family in the Caryophyllales order, comprises the *Nepenthes* genus, which has modified leaf trap characteristics. Although most *Nepenthes* species have unique morphologies, their vegetative stages are identical, making identification based on morphology difficult. DNA barcoding is seen as a potential tool for plant identification, with small DNA segments amplified for species identification. In this study, three barcode loci; ribulose-bisphosphate carboxylase (*rbc*L), intergenic spacer 1 (ITS1) and intergenic spacer 2 (ITS2) and the usefulness of the ITS1 and ITS2 secondary structure for the molecular identification of *Nepenthes* species were investigated. An analysis of barcodes was conducted using BLASTn, pairwise genetic distance and diversity, followed by secondary structure prediction. The findings reveal that PCR and sequencing were both 100% successful. The present study showed the successful amplification of all targeted DNA barcodes at different sizes. Among the three barcodes, *rbc*L was the least efficient as a DNA barcode compared to ITS1 and ITS2. The ITS1 nucleotide analysis revealed that the ITS1 barcode had more variations compared to ITS2. The mean genetic distance (K2P) between them was higher for interspecies compared to intraspecies. The results showed that the DNA barcoding gap existed among *Nepenthes* species, and differences in the secondary structure distinguish the *Nepenthes*. The secondary structure generated in this study was found to successfully discriminate between the *Nepenthes* species, leading to enhanced resolutions.

## 1. Introduction

Among the pitcher plants, the Nepentheceae family is the largest, with 120 species that produce specialized cup-shaped pitchers that attract small insects and kill them through digestive enzymes [1]. In Malaysia, there are eleven species of *Nepenthes* known to exist, including *Nepenthes gracillis, Nepenthes ampullaria* and *Nepenthes rafflesiana* [2,3]. According to previous reports, taxonomic classification of *Nepenthes* has been based on morphological characteristics such as shape, color, size and ornamentation [1,2,4,5,6,7]. Although this is common for *Nepenthes*, it frequently causes confusion because of characters that can be hard to find [8]. For example, taxonomic confusion in *Nepenthes* has been reported when *N. pilosa* was confused with *N. chaniana*, *N. talangensis* with *N. bongso,* and *N. lamii* with *N. vieillardii* [3,7]. Human interest in *Nepenthes* extends beyond these plants’ decorative nature to their therapeutic benefits, in addition to their uniqueness and beauty. The *Nepenthes* plant has traditionally been employed in traditional medicine to control the menstrual cycle, facilitate childbirth, reduce asthma, cure eye inflammation, and treat stomach ulcers, jaundice, high blood pressure, indigestion, and dysentery, and has been used as an astringent [4,9,10,11]. The “Jakun” community believes that the decoction of *N. ampullaria* can ease asthma attacks, while the stem can treat malaria [12,13]. Other pitcher plants have also been used in traditional delicacies [14]. Due to their purported health benefits, various studies have revealed the therapeutic value of plants employed in traditional herbal therapy.

Since their development, DNA barcodes have been viewed as the most promising approach to resolving this taxonomic issue [15]. Previous simulation studies for *Nepenthes* species using NCBI GenBank collection data showed that a single locus ITS or one coupled with plastid regions (matK) exhibited the best species discrimination with distinct barcoding gaps [16]. According to the previous literature, *trn*L and ITS DNA barcodes can be used to distinguish between the *Nepenthes* species based on their geographical origin area [17]. Meanwhile, other studies concluded that RNA secondary structure prediction is an advanced tool for species discrimination [18], and the integration of secondary structure information in species identification can significantly improve its accuracy for other plant species [18,19,20]. However, there are not many reports on how ITS1 and ITS2 secondary structure predictions can be used together to differentiate between species. In our study, we aimed to investigate the efficacy of *rbc*L, ITS2 and ITS1 DNA barcodes in combination with ITS1 and ITS2 secondary structure predictions in distinguishing between three *Nepenthes* species.

## 2. Materials and Methods

### 2.1. Sample Collection

The three *Nepenthes* species used in this study (*N. ampullaria*, *N. gracilis* and *N. rafflesiana*) were collected from a sampling trip in Gunung Janing, Kampung Peta, Endau-Rompin National Park, Johor, Malaysia (GPS location; 2.529908870220549, 103.41185691378324), and identified by Assoc Prof. Dr Alona Cuevas Linatoc. The geographical location from which the three *Nepenthes* samples were collected is presented in Figure 1. The collected *Nepenthes* species were wrapped with aluminum foil and kept in an ice box prior to extraction at the laboratory. The plant samples were cut and kept at −80 °C prior to extraction with designated labels of NA, NR and NG for *N. ampullaria, N. rafflesiana* and *N. gracilis,* respectively.

### 2.2. Genomic DNA Extraction and PCR Amplification

For genomic DNA extraction, frozen plant samples were first thawed before proceeding with the extraction using the commercial kit NucleoSpin Plant II (Macherey-Nagel, Duren, Germany), following the manufacturer’s instructions. Approximately 100 mg of the plant sample was ground with liquid nitrogen to produce a fine powder using mortar and pestle. The plant samples were subjected to isolation steps as detailed in the protocol. Then the genomic DNA (gDNA) obtained was quantified using a Nanodrop machine (Eppendorf, Hamburg, Germany) and checked using 1% (*w*/*v*) agarose gel electrophoresis. The integrity and quality of the gDNA obtained was verified with a Gel Documentation System (BioRad). Then, the good quality gDNAwas used as a DNA template for PCR amplification. The PCRs for *rbc*L, ITS1 and ITS2 were conducted separately in 25 µL of the total reaction volume containing 12.5 µL of 2X PrimeSTAR^®^ Max DNA Polymerase, 10 ng of gDNA for *N. gracilis, N. ampullaria* and *N. rafflesiana,* 0.625 µM of each primer set and topped up with sterile nano-pure water. The PCR was conducted using the Mastercycler^®^ nexus gradient (Eppendorf AG, Hamburg, Germany) through three different PCR profiles according to the DNA barcodes (Table 1). The PCR products were visualized on 2% (*w*/*v*) agarose gel electrophoresis in 1X TAE buffer and later sent for sequencing using Sanger sequencing at Apical Scientific Sdn. Bhd. Sequencing was performed for both the forward and reverse directions using the same primers as those used in PCR. The details of the primers are shown in Table 2.

### 2.3. Bioinformatics Analysis and Phylogenetic Tree

The forward and reverse sequences of the amplicon obtained from *rbc*L, ITS1 and ITS2 primers were edited using BioEdit software. Each generated consensus sequence of the forward and reverse sequences was submitted to the Basic Local Alignment Search Tool (BLAST) of the National Centre for Biotechnology Information (NCBI) for a homology search (https://blast.ncbi.nlm.nih.gov/Blast.cgi; accessed on 24 December 2022). The GenBank accession number for the generated barcode sequences were obtained after the sequences were submitted to GenBank via BankIt for *rbc*L, (https://submit.ncbi.nlm.nih.gov/about/bankit/; accessed on 24 December 2022) and the submission portal of NCBI for ITS1 and ITS2 (https://submit.ncbi.nlm.nih.gov/about/genbank/; accessed on 24 December 2022).

The BLASTn results were selected by determining the sequences with the maximum similarity score and lowest E value. The generated sequence of each barcode in the present study and its similarity was recorded as a percentage. Multiple sequence alignment was performed using Jalview v2.11.1.0 with all the obtained sequences. The phylogenetic analysis was performed following the neighbor-joining (NJ) tree and minimum evolution method with the 1000 *“Boostrap phylogeny”* test method using MEGAX software. The DNA best-fit substitution model for each dataset (Table 3) was determined prior to the NJ tree construction using MEGAX [23]. An outlier, *Dionaea muscipula* (accession number: AB072558.1), was selected in the NJ analysis to verify the identification of the three *Nepenthes* under study. Sequence divergences were calculated using the Kimura-2-parameter (K2P) distance model [24]. The calculation of the sequence divergences was implemented in MEGAX [23]. From the sequence divergence data, the extent of DNA barcoding gap/overlap was then explored as is typical for barcoding studies [25].

### 2.4. Secondary Structure Predictions

The DNA barcodes of the three *Nepenthes* were generated using a Bio-Rad DNA barcode generator (http://biorad-ads.com/DNABarcodeWeb/; accessed on 28 November 2022). To complement the tree-based methods, RNA secondary structure predictions were performed using the nucleotide sequences from ITS1 and ITS2 primers for the identification of the best potential barcodes using the rRNA database *RNAfold* WebServer v2.4.18 (http://rna.tbi.univie.ac.at//cgi-bin/RNAWebSuite/RNAfold.cgi; accessed on 28 November 2022). The results of RNA secondary prediction enhanced the resolution of the DNA barcodes.

## 3. Results

### 3.1. Amplification, Sequencing, Multiple Sequence Alignment and Species Identification

The DNA barcode primers, *rbc*L, ITS1 and ITS2, produced amplicons of 599 bp, 300–400 bp and 300–500 bp, respectively. In analyzing the sequences, *rbc*L was found to exhibit the most extensive sequence length (502–509 bp), followed by ITS1 (219–327 bp) and ITS2 (241–250 bp). All the sequences were submitted to GenBank, and the accession numbers were obtained. The top BLASTn score for the species identification of all three *Nepenthes* species is presented in Table 4. Using the BLASTn tool, the three *Nepenthes* species were identified as different *Nepenthes* species at different barcode regions of *rbcL, ITS1* and *ITS2*. In the BLAST search, *rbc*L and ITS2 genes identified all three *Nepenthes* species as *N. mirabilis* and *N. gracilis,* respectively. However, the barcode did not discriminate between the three *Nepenthes* species. Meanwhile, the ITS1 barcode identified the specimen NGits1 as *N. ventricosa*, while the specimens NAits1 and NRits1 were identified as *Nepenthes x intermedia*.

Multiple sequence alignment (MSA) of the DNA barcode sequences (Figure 2) was used to determine that *rbc*L had the largest alignment, followed by ITS1 and ITS2 (Figure 2a). The alignment obtained for *rbc*L showed the highest similarities (100%) among the three *Nepenthes* species, whereas ITS2 showed a slight difference in *N. ampullaria, (NAits2)* with a ten-nucleotide difference between *N. gracillis* (NGits2) and *N. rafflesiana* (NRits2) (Figure 2c) at nucleotides 15, 31, 77, 80, 85, 98, 110, 115, 149 and 154. In contrast, ITS1 showed the highest difference between the three *Nepenthes* species (Figure 2b) based on the length and nucleotide composition of each sequence. Indisputably, among the three ITS1 sequences, NGits1 showed the highest variation followed by NAits1 and NRits1. Variation between NAits1 and NRits1 were based on the difference in sequence length, as NRits1 is fourteen nucleotides longer than NAits1 with a two-base difference at nucleotides 44 and 73. This corresponds with the ATGC percentage of the sequences showing that *rbc*L had the lowest GC content (44.6–45%), followed by ITS2 (63.1–66.5%) and ITS1 (66.1 to 68.5%) (Table 5). Nucleotide composition analysis also allowed us to conclude that sequences amplified using the *rbc*L barcode have the least variability since the AT and GC percentages were almost the same. Meanwhile, both ITS1 and ITS2 sequences show variations in AT and GC content, indicating more variability.

All the sequences generated from the amplification of *rbc*L, ITS1 and ITS2 barcodes were successfully deposited into the GenBank database. The accession number for each barcode is as follows: *rbc*L: OP534746 (*N. ampullaria*), OP534747 (*N. rafflesiana*), OP534748 (*N. gracilis*); ITS1: OQ123732 (*N. ampullaria*), OQ123724 (*N. rafflesiana*), OQ123725 (*N. gracilis*); and ITS2: OQ123720 (*N. ampullaria*), OQ123721 (*N. rafflesiana*), OQ123722 (*N. gracilis*).

### 3.2. Phylogenetic Studies, Intraspecific Variation, Interspecific Divergence and DNA Barcoding Gap

Phylogenetic analysis using an NJ tree in the respective DNA best-fit substitution model, as shown in Table 3 with bootstrap-1000 of the three *Nepenthes* species, showed a high similarity in the BLAST search to the available sequences in the NCBI database (Figure 3). Although there was no significant variation between the generated barcode sequences from the three *Nepenthes* species when observed through multiple sequence alignment, phylogenetic analysis revealed further differences between the three *Nepenthes* under study (NA, NR and NG), especially when ITS1 and ITS2 were considered. The NJ tree of *rbc*L showed that the three species studied were in the same clade as the other species, *N. mirabilis* and *N. alata* (Figure 3a), with the same node scores of 26% for *N. gracilis, N. ampullaria* and *N. rafflesiana*, respectively. Meanwhile, NJ analysis of ITS1 sequences showed that the three species studied could be classified into different clades; NAits1 is in a clade with NRits1 with a node score of 78%, while NG was classified into an individual clade (Figure 3b). Further phylogenetic characterization using ITS2 revealed that NRits2 and NGits2 are related to *N. rafflesiana* (accession number: HM204904.1) since they were grouped into the same clade with node scores of 73% and 77%, respectively (Figure 3c). The NAits2 appeared individually at a clade near NRits2, NGits2 and a clade of another *N. ampullaria* species.

Previous reports on the intraspecific and interspecific divergence among species are useful for assessing the potential of DNA barcodes [25,26,27]. Based on the neighbor-joining (NJ) tree of K2P distances, taxa or groups were organized to calculate the intraspecific variations and interspecific divergences among them. The *Nepenthes* species under study showed unique clades and within-species sequence divergence between 0 and 4% in *rbc*L, 0 and 5.9% in ITS2, and 0 and 26.9% in ITS1, whereas divergence between species ranged from 0 to 1% in *rbc*L, 1 to 8% in ITS2, and 1.5 to 44% in ITS1 (Figure 4a–c). The results indicate that the interspecific divergence was distinctly higher for the interspecific divergence distance for ITS2 and ITS1, and that, consequently, a clear DNA barcode gap was present (Figure 4b,c). In addition, the results obtained with ABGD analysis are consistent with the results obtained with MEGAX, evidencing DNA barcode gaps.

### 3.3. DNA Barcodes, ITS1 and ITS2 Secondary Structure Predictions

Figure 5 and Figure 6 show the DNA barcodes and ITS1 and ITS2 secondary structure predictions based on minimum free energy (MFE). The highest MFE for NAits1, NGits1 and NRits1 was observed at 100–105 bp (Figure 5a), 120–125 bp (Figure 5b) and 115–120 bp (Figure 5c), respectively. Meanwhile, the highest MFE for both NAits2 and NRits2 was observed at 45–50 bp (Figure 6a,c), while NGits2 recorded the highest MFE at 120–125 bp (Figure 6b). DNA barcode sequences derived from ITS1 showed variations among the three *Nepenthes* species (Figure 5d–f). NGits1 exhibited the highest barcode length (328 bp), followed by NRits1 (233 bp) and NAits1 (220 bp).

Similarly, ITS1 secondary structure predictions show further variation between the three Nepenthes species. The predicted ITS1 secondary structures of the three Nepenthes showed that each structure had a central ring with different helical orientations (Figure 4g–i). Generally, the secondary structures of NAits1 and NRits1 showed a tighter configuration and pattern similarity despite having a different number of loops on the central ring and along the helices. NGits1, on the other hand, exhibited a more complex structure compared to NAits1 and NRits1, with a bigger central ring, different helix orientations and lengths, varied loop numbers and variation in angles from the spiral. The loop number, position, size and angle from the centroid were distinguishable in all three Nepenthes species.

The predicted ITS2 secondary structure provides additional information on the differences between the three Nepenthes species by representing three slightly different structures with a central ring with different helical orientations (Figure 5g–i). In general, the predicted ITS2 secondary structures had a more uniform pattern compared to the predicted ITS1 secondary structures. All three predicted ITS2 secondary structures shared the same central and backbone pattern, but the loop number, position size and angle from the centroid were still distinguishable in all three Nepenthes species. Consequently, in ITS1′s secondary structure, NAits1 showed the most obvious pattern difference, with a larger central ring compared to NRits1 and NGits1. Based on the results, both ITS1 and ITS2 secondary structure predictions provide deeper insights into the differences between the three Nepenthes species, allowing each species to be identified as unique species and guiding comparative sequence analysis. In addition, the prediction of secondary structures can further assist in the design of species-specific RNA molecules.

## 4. Discussion

DNA barcoding is used in many plant biodiversity studies to identify species [28,29], discover new taxa, conserve species and enable studies of plant ecology by constructing phylogenetic trees [30]. In this study, ITS1 and ITS2 showed a greater ability for species discrimination than *rbc*L, especially with regard to secondary structure predictions. Given the universality of the *rbc*L gene, it has been proposed as a universal barcode fragment due to its ease of amplification and comparison [31]. It is widely used for phylogenetic analysis of families and subclasses in various seed plant groups [32]. However, the limitation of the *rbc*L barcode is its inability to discriminate between organisms up to the species level. Although Nepenthes does not belong to the seed plant group, the present study showed that *rbc*L could not classify the three *Nepenthes* up to the species level, indicating a poor ability to distinguish between species. This was also supported by nucleotide analysis and NJ, which only reached the genus level. This finding is consistent with earlier research showing that diversity in the *rbc*L sequence is scarce at the species level [26,33,34,35] and reduces its ability to be used to discriminate between species. The high universality and low resolution of *rbc*L indicates its inefficiency as a DNA barcode, which is also suggested for other plant species, such as *Acacia* [36].

The ITS region is one of the most commonly used barcodes in genus- and species-level phylogenetic analyses in eukaryotes [37,38]. A previous study had shown the application of DNA barcodes for three *Nepenthes* species (*N. ampullaria, N. rafflesiana* and *N. gracilis*), which enabled them to differentiate between them based on their geographical area of origin. The phylogenetic analysis was based on *trn*L and ITS barcodes but lacked information on the effectiveness of the two DNA barcodes in distinguishing the *Nepenthes* spp. from each other [17]. Additionally, a previous study reported that the *trn*L intron does not represent the best choice for characterizing plant species and for phylogenetic studies among closely related species [39]. The present study involved plant samples from very closely related species; thus, we made use of *rbc*L*,* ITS1 and ITS2 to evaluate the species variation. Our findings indicate that the *Nepenthes* species were grouped at different clades when using ITS1 (Figure 3b), with *N. gracilis* appearing individually in one clade while *N. ampullaria* and *N. rafflesiana* were grouped into another clade. These results show that ITS1 can be used to discriminate between the three *Nepenthes* species but did not effectively identify the plant samples up to the species level. In contrast, the ITS2 phylogenetic tree revealed that these three species were closely clustered together with the sequences of *N. ampullaria, N. rafflesiana* and *N. gracilis* chosen from the BLASTn search, although they did not appear in the same clade (Figure 3c).

Overall, NJ analysis showed that the phylogenetic tree method poorly distinguished between the Nepenthes species compared to distance-based analysis. A previous study showed that barcode gaps act as typical barcode data characterized by having differences between intraspecific diversity and interspecific diversity [25]. However, this was not a general feature in all groups. In this study, the histogram and ranked pairwise (K2P) distance analyzed using ABGD programs showed that the “barcoding gap” between levels of intraspecific variation and interspecific divergence did exist for the analysis of *rbc*L, ITS2 and ITS1 (Figure 4a–c). Compared to the ITS1 barcoding gap, ITS2 has a wider barcoding gap number. In the present study, phylogenetic analysis did not ambiguously identify the species, but the predicted ITS1 and ITS2 secondary structures revealed the uniqueness of the generated DNA barcode sequences. According to a previous study, the features of the ITS2 region are conserved, which in turn provides molecular and morphological features that could be used to improve the resolution of species identification [40]. The usefulness of ITS2 sequences and the ITS2 secondary structure as genetic markers has been reported for several medicinal plant species [20,41,42]. Regarding the ITS2 secondary structure, the conservative standard structural model contained the feature of a “loop” with four “helices.” Among the four “helices,” helix I and helix IV had the most variations, helix II and helix III were more conserved, helix II was shorter with a pyrimidine–pyrimidine mismatch “bridge,” while helix III was the longest with many branches [30,43,44].

In our study, prediction of secondary structures in all the *Nepenthes* species under study revealed diverse secondary structures with distinguishable loop numbers, positions and elevations from the centroid. This information can be applied to design species-specific primers for identifying genotypes. Prediction of secondary structure differentiation indicated variation among RNA molecules in all species when using either ITS1 or ITS2. The alignment of the primary nucleic acid sequences could be optimized and modified using the secondary ITS2 information, contributing to the accuracy and robustness of the phylogeny [45]. Previous research had shown that genetic structure uniqueness at the conserved nuclear region could be useful for developing species-specific primers, as reported in a previous study [18]. Meanwhile, little information was available on the function and secondary structure of ITS1, leaving a gap in the knowledge on whether ITS1 secondary structure prediction could enhance species identification. Consequently, the region of ITS1 may play an important role in 18SrRNA maturation [43]. A few reports suggest DNA barcodes can be used to efficiently discriminate between *Nepenthes* species [16,17,46]. However, until now, no study has reported ITS1 and ITS2 secondary structure predictions in relation to species discrimination in *Nepenthes*. Our study revealed significant variations in both ITS1 and ITS2 secondary structure predictions that enhanced species discrimination between the three *Nepenthes* species under study.

## 5. Conclusions

This study describes the efficiency of DNA barcode genes from *rbc*L, ITS1 and ITS2 in differentiating between three *Nepenthes* along with secondary structure predictions. Although the *rbc*L gene in the chloroplast–plastid region might be easily amplified, it has a poor species identification and discrimination ability. On the other hand, the incorporation of secondary structures in the nuclear ribosomal genes of ITS1 and ITS2 may serve as a trustworthy tool in species identification and designing species-specific primers. The findings of this investigation provide clarity on the scientific foundations for species identification, genetic preservation and the secure use of this significant species of medicinal plant. The employment of DNA barcode technologies for species delimitation in commercially and medicinally important plant species may be possible.

## Figures and Tables

**Figure 1 genes-14-00697-f001:**
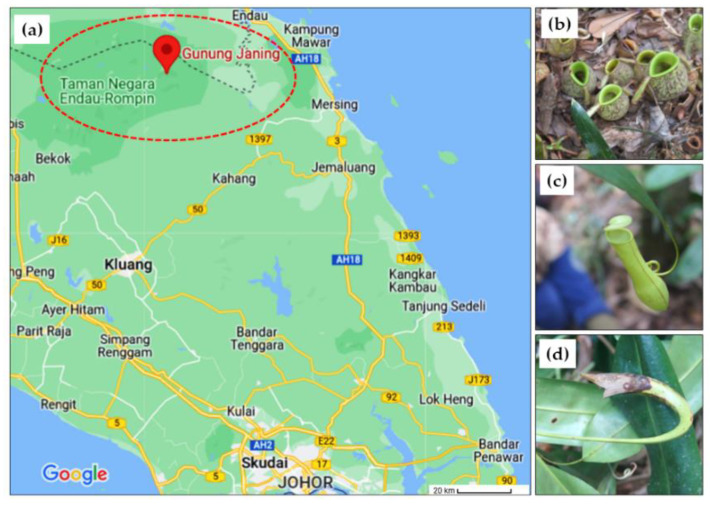
The geographical location (**a**) of the *Nepenthes* collected from Bukit Janing, Kampung Peta, Endau-Rompin National Park, Johor, Malaysia (GPS location; 2.529908870220549, 103.41185691378324). *Nepenthes* species collected: (**b**) *Nepenthes ampullaria*, (**c**) *Nepenthes gracilis* and (**d**) *Nepenthes rafflesiana*.

**Figure 2 genes-14-00697-f002:**
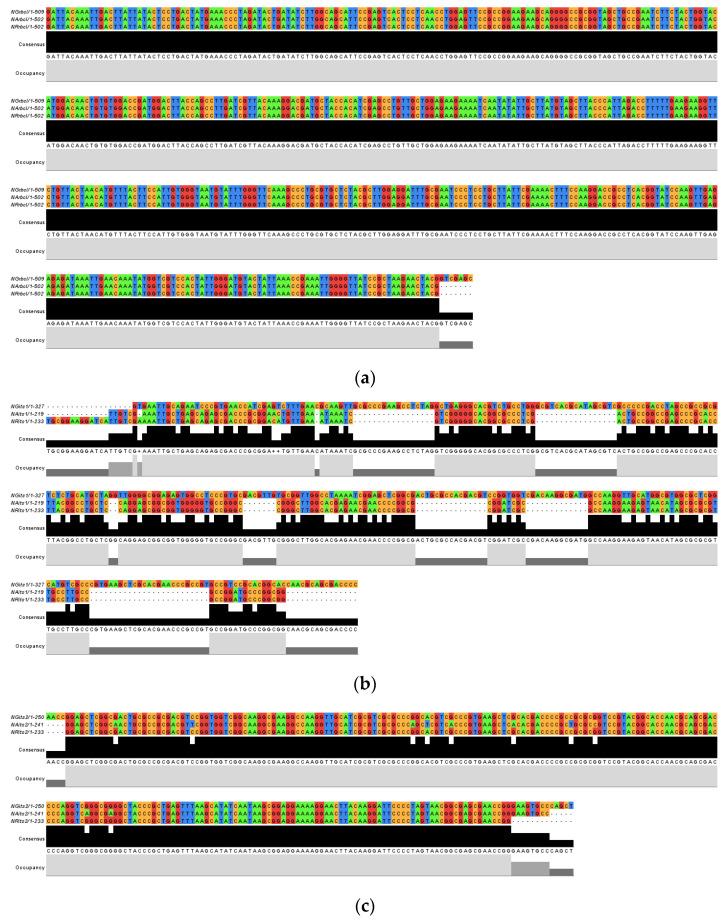
Multiple sequence alignment of *rbc*L (**a**), ITS1 (**b**) and ITS2 (**c**) barcodes.

**Figure 3 genes-14-00697-f003:**
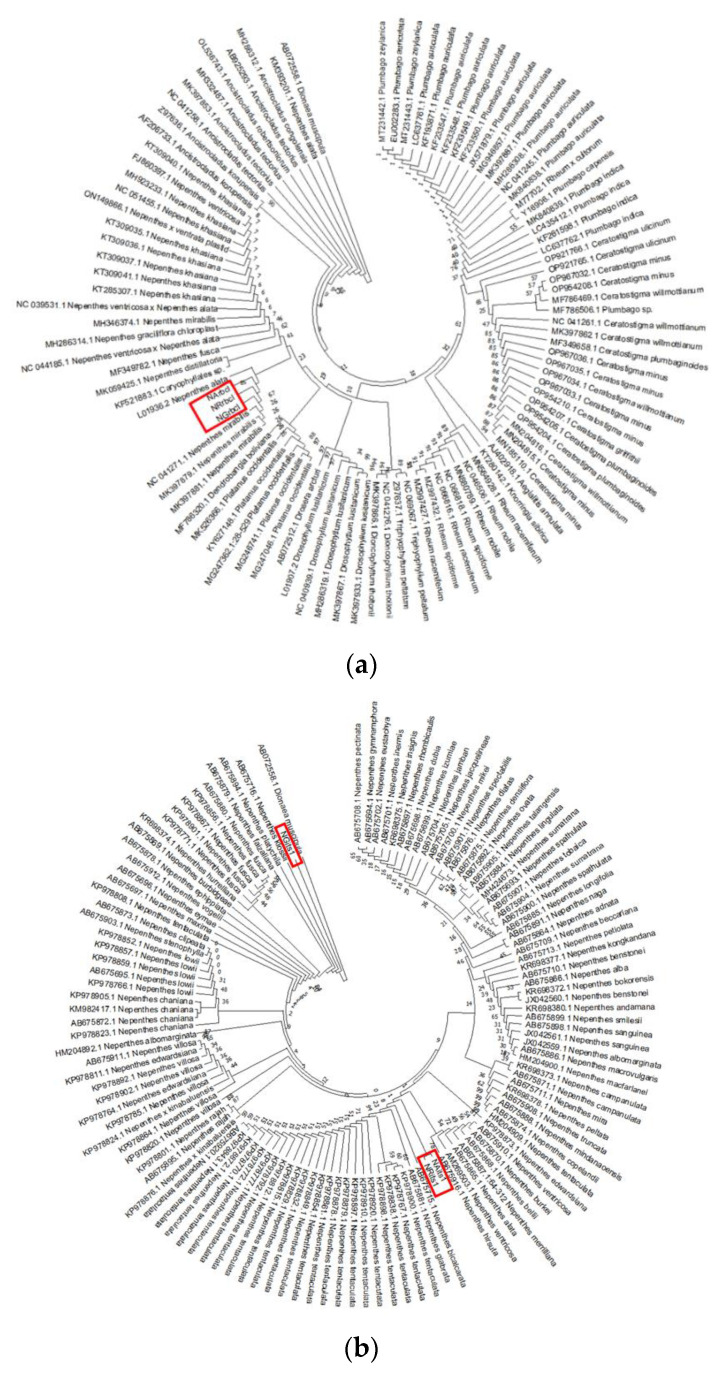
Neighbor joining tree of *rbc*L (**a**), ITS1 (**b**) and ITS2 (**c**) barcode primers depicting the phylogenetic analysis among *N. ampullaria, N. gracilis,* and *N. rafflesiana* (indicated in the red boxes).

**Figure 4 genes-14-00697-f004:**
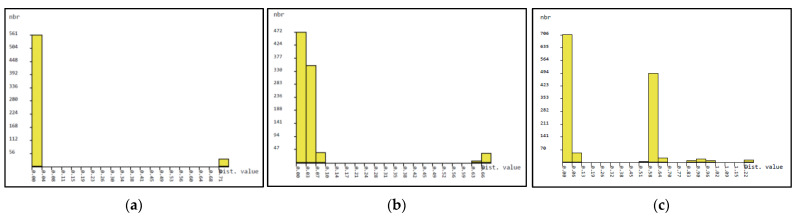
Barcode gap analysis of *N. gracilis*, *N. ampullaria*, and *N. rafflesiana* generated by Automatic Barcode Discovery Gap Discovery (https://bioinfo.mnhn.fr/abi/public/abgd/abgdweb.html, accessed on 23 January 2023) [25] for (**a**) rbcL, (**b**) ITS2 and (**c**) ITS1. Distributions of K2P distances and between each pair of specimens for the histogram of distance.

**Figure 5 genes-14-00697-f005:**
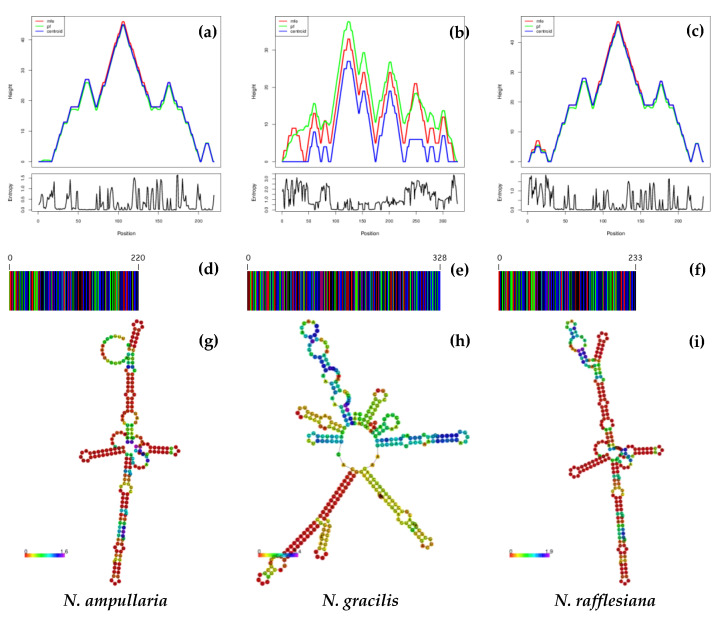
The variations observed in the predicted minimum free energy (MFE) (**a**–**c**), DNA barcodes (**d**–**f**), and secondary structures of the ITS1 region for the three *Nepenthes* species (**g–i**).

**Figure 6 genes-14-00697-f006:**
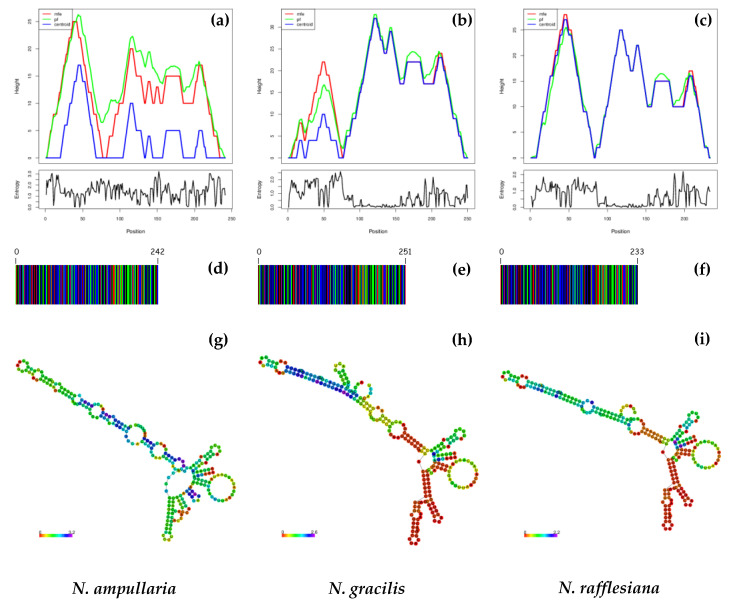
The variations observed in the predicted minimum free energy (MFE) (**a**–**c**), DNA barcodes (**d**–**f**), and secondary structures of the ITS2 region for the three *Nepenthes* species (**g–i**).

**Table 1 genes-14-00697-t001:** Polymerase Chain Reaction (PCR) profiles used in this study.

Steps	No. of Cycles	*rbc*L ^1^		ITS1 ^2^		ITS2 ^3^	
		Temperature (°C)	Duration (s)	Temperature (°C)	Duration (s)	Temperature (°C)	Duration (s)
Denaturation	35	98	10	98	10	98	10
Annealing		53	15	55	15	55	10
Elongation		72	20	72	40	72	30
Hold	1	4	∞	4	∞	4	∞

^1^ ribulose-bisphosphate carboxylase (*rbc*L). ^2^ intergenic spacer 1 (ITS1).^3^ intergenic spacer 2 (ITS2)

**Table 2 genes-14-00697-t002:** List of primers used in this study.

Region	Primer Name	Primer Sequence (5′ to 3′)	Primer Length (bp)	Reference
*rbc*L	Rbcla_fwd	ATGTCACCACAAACAGAGACTAAAGC	26	[21]
	Rbclb_rvs	GTAAAATCAAGTCCACCRCG	20	
ITS1	ITS1_fwd	GGAAGGAGAAGTCGTAACAAGG	22	[21]
	ITS1_rvs	AGATATCCGTTGCCGAGAGT	20	
ITS2	ITS2_fwd	GGGGCGGATATTGGCCTCCCCTTG	24	[22]
	ITS2_rvs	GACGCTTCTCCAGACTACAAT	21	

**Table 3 genes-14-00697-t003:** DNA best-fit substitution model depicted from MEGAX software for the three DNA barcodes.

Barcode Genes	DNA Best-Fit Substitution Model
*rbc*L	Kimura-2-parameter
ITS1	Tamura-3-parameter
ITS2	Kimura-2-parameter + γ distribution

**Table 4 genes-14-00697-t004:** Molecular identification of *Nepenthes* species using *rbc*L, ITS1 and ITS2 barcode genes.

Barcode Genes	Sample ID	Scientific Name	Accession Number	E Value	Query Coverage (%)	Percent Identity (%)
*rbc*L	NA*rbc*L	*N. mirabilis*	NC_041271.1	0.0	100	100
	NG*rbc*L	*N. mirabilis*	NC_041271.1	0.0	100	100
	NR*rbc*L	*N. mirabilis*	NC_041271.1	0.0	100	100
ITS1	NAits1	*Nepenthes x intermedia*	HM204899.1	1e-103	100	98.64
	NGits1	*N. ventricosa*	AB675910.1	2e-168	100	100
	NRits1	*Nepenthes x intermedia*	HM204899.1	2e-116	100	100
ITS2	NAits2	*N. gracilis*	AB675882.1	4e-104	100	95.85
	NGits2	*N. gracilis*	AB675882.1	2e-122	100	99.20
	NRits2	*N. gracilis*	AB675882.1	5e-113	100	99.14

**Table 5 genes-14-00697-t005:** The average AT and GC percentage nucleotide composition of *N. gracilis, N. ampullaria* and *N. rafflesiana* based on *rbc*L, ITS2 and ITS1 DNA barcodes.

DNA Barcode	Species	AT (bp)	GC (bp)	Total (bp)	AT (%)	GC (%)
*rbc*L	*N. gracilis*	280	229	509	55	45
	*N. ampullaria*	278	224	502	55.4	44.6
	*N. rafflesiana*	278	224	502	55.4	44.6
ITS2	*N. gracilis*	85	114	250	34.0	66.0
	*N. ampullaria*	89	152	241	36.9	63.1
	*N. rafflesiana*	78	155	233	33.5	66.5
ITS1	*N. gracilis*	111	216	327	33.9	66.1
	*N. ampullaria*	69	150	219	31.5	68.5
	*N. rafflesiana*	76	157	233	32.6	67.4

## Data Availability

Not applicable.

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
