# Peer review of "DNA Barcoding, Phylogenetic Analysis and Secondary Structure Predictions of Nepenthes ampullaria, Nepenthes gracilis and Nepenthes rafflesiana"

_genes, 2023, doi:10.3390/genes14030697_

Round 1
Reviewer 1 Report
The article is well written in all its paragraphs. The introduction is clear and immediately places the reader in the scenario of this research. The methods used are well described in the dedicated paragraph. The results obtained support the conclusions of the authors. Unfortunately I have to pay attention to the very little originality of the work both for the topic and for the experimental approach used. Molecular markers such as those used here are very valid, but using only the results derived from the variability between the sequences is very little. The work could clearly become more interesting if the authors build a panel of snp's identifying each species on the basis of these results. As it is, it is a good paper but very uninteresting
Reviewer 2 Report
This paper offers a contribution to the field of plant identification. The English needs to be improved in many parts and there are several errors and inaccuracies, including the use of non italicised names for the genus Nepenthes and for many other genus names in the literature. The figures resolution is too poor, and the authors should include a map of the larger region showing the sampling site's location in Malesia. Additionally, the results are not novel, as another study has already differentiated between the three species using trnL and ITS (ref 6). Additionally, it would be helpful if the authors could clarify their order of citation inclusion in their text, as the first citation is number 26 and the second is number 8. The authors should explain why they did not use trnL in their study, as well as address any taxonomic errors, such as the statement that Nepentheceae family is the largest genus (?). Furthermore, the sampling process is unclear, making it difficult to determine for example how the authors could be sure the three samples belonged to the three species and why they chose those tree locations. Finally, there are several inaccuracies and omissions in the references. We kindly suggest the authors revise the paper to address these issues before submitting it again.
Round 2
Reviewer 1 Report
The paper can be accepted without any further changes.